# Pre- and Post-Operative Hamstring Autograft ACL Reconstruction Isokinetic Knee Strength Assessments of Recreational Athletes

**DOI:** 10.3390/jcm12010063

**Published:** 2022-12-21

**Authors:** Nizamettin Güzel, Ali Kerim Yılmaz, Ahmet Serhat Genç, Emre Karaduman, Lokman Kehribar

**Affiliations:** 1Department of Orthopedics and Traumatology, Samsun Training and Research Hospital, Samsun 55090, Turkey; 2Faculty of Yasar Dogu Sport Sciences, Ondokuz Mayıs University, Samsun 55270, Turkey; 3Department of Orthopedics and Traumatology, Samsun University, Samsun 55090, Turkey

**Keywords:** anterior cruciate ligament reconstruction, athletes, isokinetic assessments, return to sport, semitendinosus/gracilis

## Abstract

Background and Objectives: Anterior cruciate ligament (ACL) injuries are common injuries with a high incidence among people with high physical activity levels. Therefore, ACL reconstruction (ACLR) is one of the most common surgical procedures performed in sports medicine. This study aims to compare the pre- and 6-month post-operative isokinetic knee strengths in healthy (HK) and ACL knees of patients who underwent semitendinous/gracilis (ST/G) ACLR. Materials and Methods: A retrospective cohort of 21 recreational athletes who underwent ST/G ACLR by the same surgeon were evaluated. The pre- and 6-month post-operative isokinetic knee extension (Ex) and flexion (Flx) strengths of the HK and ACLR patients were evaluated in a series consisting of three different angular velocities (60, 180 and 240°/s). Of all the findings, peak torque (PT) and hamstring/quadriceps (H/Q) parameters were evaluated. Results: There was a significant improvement in post-operative Lysholm, Tegner and IKDC scores compared to pre-operative scores (*p* < 0.05). There were significant differences in pre-operative and post-operative knee Ex and Flx strengths at angular velocities of 60°, 180° and 240°/s in both the ACLR and HK groups (*p* < 0.001). There was no significance at 240°/s Flx for ACLR (*p* > 0.05). As for H/Q ratios, there was a significant difference between pre- and post-operative values only at 60°/s angular velocity in both ACLR and HC (*p* < 0.005). Conclusions: The pre-operative and 6-month post-operative results of the ST/G ACLR showed that there was a high level of recovery, particularly in quadriceps strength, while the increase in strength was less in the hamstring. The significance observed at 60°/s in H/Q ratios was within normal ranges. It can be argued that the ST/G ACLR method is feasible for people with high physical activity levels and for athletes.

## 1. Introduction

Anterior cruciate ligament (ACL) injuries have a significant impact on knee stability and motor control mechanisms of the knee [1]. ACL rupture, which is frequently observed in people with high physical activity levels, has a high incidence and can cause cartilage and meniscus damage and, accordingly, osteoarthritis if not treated surgically [2]. The mechanical, proprioceptive and efferent neuromuscular disorders resulting from ACL rupture can cause muscle deficits and significant loss in muscle strength. In addition, hamstring strain injuries, which are frequently encountered in people with high levels of physical activity, such as ACL injuries, can often occur during sharp turns, hitting the ball and sprinting, especially after ACL injuries [3,4]. The ACL is of great importance particularly for individuals with high levels of physical activity, as it adjusts the stiffness of quadriceps (Q) and hamstring (H) muscles, which are the agonist-antagonist structure of the knee. It also enables safe reciprocal movements and performs a proprioreceptive function [5]. From an athletic point of view, ACL rupture often occurs as a result of sudden movements [6]. In view of all these aspects, ACL reconstruction (ACLR) is recommended by orthopedists and sports physicians for individuals with high physical activity levels. Although different grafting methods such as quadriceps tendon (QTA) and patellar tendon (PTA) are used for ACLR, one of the most commonly used methods is the hamstring autograft, which is performed using semitendinosus/gracilis (ST/G) tendons [7].

ACLR aims to restore joint function and stability. On the other hand, rehabilitation processes aim to restore neuromuscular losses, particularly of Q and H muscles [8,9]. For the athletic population and individuals with high physical activity levels, a decent rehabilitation and strengthening process after ACLR is crucial in terms of time to return to sport (RTS). Although researchers use different methods to determine post-operative RTS times, the most objective data regarding strength are provided by isokinetic dynamometers [10,11,12,13]. With isokinetic dynamometers, it is possible to comment on RTS using Flx/Ex or H/Q ratios which occur as a result of strength produced by Q and H muscles in the extension (Ex) and flexion (Flx) phases of the knee. They are also useful in giving an idea of when the knee can resume physical activity regarding strength. The imbalances in the H/Q ratios used after ACLR can be defined as asymmetrical strength for the lower extremity. This ratio increases as the angular velocity increases in isokinetic dynamometers and can vary between 50 and 80% [14,15]. For an angular velocity of 60°, a H/Q ratio of 2/3 can be considered normal [16].

Studies have reported that knee strength in the pre-operative ACLR process is directly related to post-operative knee functions, strength and recovery [17]. Therefore, it is assumed that exercise and rehabilitation processes applied not only after ACLR but also in the pre-operative period affect the post-operative findings positively. However, although there is sufficient information in the literature for post-operative rehabilitation processes and exercise programs, studies evaluating the strength development in the pre-operative period are still limited [18,19,20]. Researchers have reported that strength losses before ACLR could not be compensated for up to 2 years post-operation in certain cases [21]. Therefore, it is evident that strength values before and after ACLR are of great importance, particularly in the athletic population and individuals with high physical activity levels for whom RTS is crucial.

In the light of this information, the aim of the present study is to evaluate the isokinetic knee Ex and Flx strengths of individuals who are recreationally engaged in active sports before and after ACLR, and the H/Q ratios have been used to determine RTS. It is believed that this retrospective study conducted on individuals with high activity levels is important for athletic population evaluations, which are limited in the literature. The present study has two hypotheses: (1) The strength increase in the ACLR group of subjects would be higher than that of the healthy group (HK). (2) H/Q ratios would not change in the HK group as a result of rehabilitation and strengthening but would return to normal ranges in the ACLR group.

## 2. Materials and Methods

### 2.1. Experimental Approach of the Study

This study was conducted using a retrospective institutional registry of patients treated for ACL rupture at an academic medical center. Lysholm, Tegner and International Knee Documentation Committee (IKDC) scores and knee isokinetic Ex and Flx strengths at different angular velocities (60°/s, 180°/s and 240°/s) were obtained from patients pre- and 6-month post-ACLR between January 2020 and December 2021. All data collected were part of a routinely ongoing clinical procedure and were derived from medical records. The study was performed at a single center and by a single surgeon specializing in soft tissue knee injuries. In addition, all patients were referred to the same rehabilitation specialist and followed-up after surgery (for a 6-month period).

### 2.2. Patients

A total of 28 patients were included in the study but 7 patients were later excluded due to missing data (Figure 1). In this study, the data of 21 male volunteers who were doing sports recreationally in various sports branches were analyzed. The mean age was 25.6 ± 5.8 years. Inclusion criteria for this study were: (a) being 18–35 years of age, (b) doing sports actively (recreationally) for at least 5 years, (c) having undergone Semitendinosus/Gracilis (Hamstring Autograft) anterior cruciate ligament reconstruction (ACLR), and d) having no history of another neuromuscular or musculoskeletal injury or contralateral knee surgery or injury. In addition, patients who did not comply with the rehabilitation schedule, did not complete the post-operative laboratory visit or were exposed to various complications during the follow-up were excluded from the study (Figure 1). Of all the patients, 64.3% (n = 12) had a right knee injury. Detailed characteristics of the recruited patients are displayed in Table 1. This study was approved by the local ethics committee under the number “SÜKAEK-2022-10/2”.

### 2.3. Surgical Treatment (Semitendinosus/Gracilis Autograft)

In ST/G ACLR, semitendinosus and gracilis tendon autografts from the same leg are used. Both tendons are folded in half to form a four-strand graft. A closed socket is opened into the femur via the medial arthroscopic portal, and an open tunnel is opened into the tibia from the outside. Suspension fixation is used to fix the graft to the femur, and interference screw fixation is used to fix it to the tibia. The ST/G graft used in the study was anatomically at a minimum of 7, a maximum of 10 and an average of 8.3 diameters. It was observed that when the graft was folded in half it was between 12 and 13 cm on average.

### 2.4. Procedures

Lysholm, Tegner and IKDC scores and isokinetic knee strengths at different angular velocities (60°, 180° and 240°/s) were measured pre-operatively and post-operatively for both legs. A standardized pre- and post-operative protocol was used for all patients. No adverse event or complication was recorded during or after surgery (patients completing the tests, n = 21).

Patients visited the laboratory three times for routine clinical measurements after surgery. The first visit involved informing subjects about the study, obtaining anthropometric data and experiencing isokinetic measurements (familiarization). In the second visit, Lysholm, Tegner and IKDC scores and isokinetic knee strength values (pre-operative outputs) of the patients were obtained. In the third visit, Lysholm, Tegner and IKDC and isokinetic knee strength measurements were repeated (6 months after the operation) to obtain the post-operative results of the participants.

#### 2.4.1. Anthropometric Measures

Body mass (with sports clothes and without shoes) was measured to the nearest 0.01 kg using a calibrated electronic scale (SECA, Hamburg, Germany), and height (anatomical position) was measured to the nearest 0.1 cm with a stadiometer attached to the weight scale. Body mass index (BMI) was calculated by dividing body weight in kilograms by height in meters square (BMI = kg/m^2^). Patients’ general anthropometric characteristics are presented in Table 1.

#### 2.4.2. Isokinetic Strength Measurement

Knee extension and flexion strengths were measured using an isokinetic dynamometer (Humac Norm Testing and Rehabilitation System, CSMI, USA) (Figure 2). The dynmometer was adjusted in accordance with the fixed protocol (seat, dynamometer and adapter) set for knee extension and flexion strengths.

Isokinetic measurements of the patients were conducted for concentric contraction at three angular velocities (60°/s, 180°/s and 300°/s). Participants were sat upright on an adjustable dynamometer seat and each patient’s trunk, pelvis and thighs were fastened/stabilized to prevent unusual body movements. The pad on which the lower leg attachment was fixed was placed proximal to the lateral malleus. In addition, the ankle was placed on the leg stabilizer under the dynamometer seat to prevent movement in the contralateral limb. The range of motion (RoM) of the dynamometer was set between 0° (knee extension) and 90° (knee flexion). The knee joint rotation axis was identified through the lateral femoral condyle and aligned with the motor axis. Gravity correction was applied (at 90° full extension) to eliminate the antigravity effect of the limbs. The test protocol on the dynamometer included a warm-up set for each angular velocity at the same pace followed in the test set. In the warm-up set, patients performed sub-maximal 4-repeat flexion/extension movements.

Isokinetic knee Ex and Flx strength tests for both HK and ACLR groups were performed with concentric/concentric (Con/Con) contractions aligned at 60°/s (15 s din/5 reps), 180°/s (15 s din/5 reps) and 240°/s. Participants were given 1 min rest intervals between test sessions (i.e., between angular velocity changes) to minimize fatigue. Once the testing of one side was completed, there was a 5 min interval, during which the dynamometer setting was changed to adjust to the contralateral lower extremity. Measurements were first conducted on the ACLR knee. In order to ensure maximum effort, all participants were given standard verbal encouragement and were asked to apply maximum force. All tests were performed in the same order by the same researcher and at the same time of the day (10:00–14:00). The duration of the test protocol was approximately 15–20 min for each patient. The values obtained from the measurements were recorded as force-based torque values (Newton meters). In addition, H/Q ratios were recorded as percentages (%). The dynamometer was calibrated before each laboratory visit.

### 2.5. Statistical Analyses

Statistical analyses were conducted using SPSS 21 (IBM Inc., Chicago, IL, USA). Descriptive data were presented as the mean, standard deviation, median, minimum and maximum. The data were checked for normality using the Shapiro–Wilk test and were examined for kurtosis and skewness. The paired samples *t*-test was used to compare Lysholm, IKDC and Tegner outcome scores pre- and post-operation in a normal distribution, and the Wilcoxon test was used in non-normal distribution. Comparisons between the groups over time were determined using a mixed repeated-measures analysis of variance (RM ANOVA) with a Bonferroni correction for post hoc analyses. The ‘time’ for knee peak torque variables and H/Q ratios at pre- and post-operation and ‘group’ (ACL-Knee and Healthy-Knee) were considered as within-subjects and between-subjects factors at 2 × 2 time and group levels, respectively. Additionally, as a key assumption, we tested for interaction between time and group. Significance was set at *p* < 0.05, with associated 95% confidence intervals.

## 3. Results

Lysholm, IKDC and Tegner scores of the patients before and after the operation are presented in Table 2. Post-operative Lysholm (*p* = 0.001), IKDC (*p* < 0.001) and Tegner (*p* = 0.008) scores showed a significant improvement in patients (Table 2).

In Table 3, 60°/s, 180°/s and 240°/s flexion and extension knee strength averages for the ACL-Knee and Healthy-Knee groups are shown pre- and post-operation. In addition, H/Q ratios (60°, 180° and 240°) are compared for both knees pre- and post-operation. Knee extension and flexion strengths before and after surgery at angular velocities of 60° (*p* < 0.001; <0.001) 180° (*p* < 0.001; <0.001) and 240°/s (*p* < 0.001; <0.001) in the ACL-Knee group showed significant differences (respectively). While there was a difference in the H/Q ratio of 60° (*p* = 0.007) between pre- and post-operation for the ACL-Knee groups, there was no difference in H/Q ratios at angular velocities of 180° (*p* = 0.552) and 240° (*p* = 0.085). Pre- and post-operative results of 60°, 180° and 240°/s knee strengths and H/Q ratios for the Healthy-Knee group were similar to those of the ACL-Knee group results (*p* < 0.05). Furthermore, analysis of variance (RM ANOVA), by examining the two-way interaction of operated side (ACL-Knee versus Healthy-Knee) and time (post-operative versus post-operative), revealed statistically significant interaction effects for 60° (<0.001) and 240°/s (*p* = 0.047) extension and 60°/s (<0.001) flexion. Finally, ANOVA showed a significant time x group interaction effect at 60°/s H/Q (*p* = 0.001) (Table 3).

Figure 3 presents the percentage differences between the pre- and post-operative values of isokinetic knee strengths (at all angular velocities) and H/Q ratios for ACL-Knee and Healthy-Knee groups.

## 4. Discussion

Our current findings show that in the short-term post-operation after a ST/G ACLR, there is an improvement in strength for the ACLR group in both the Ex and Flx phases. However, although there was an increase in strength, a slight decrease was observed in the H/Q ratios. However, when this decrease is evaluated within the norm ranges, it is not in the risky range. When the pre- and post-operative findings for both sides were evaluated, it was seen that the ACLR group had greater strength gain than the HK group. When evaluated in terms of knee scores, it was determined that the short-term postoperative findings of the ST/G ACLR technique were directly proportional to the improvement. It was found that significant findings at different levels emerged in studies designed with prospective cohorts on subject groups who underwent ACLR using different graft types. In clinical evaluations of the hamstring autograft and bone patellar tendon bone autograft methods, researchers have found postoperative knee scores similar to our current study [10,11,19,21]. Although researchers generally found similar findings in postoperative knee scores with different types of ACLR autograft methods, no clear idea around the return to sports has emerged. Even though some of the researchers stated that the bone patellar bone tendon provides a shorter return to sport and daily physical activity time compared to the hamstring autograft [21], some argue that the hamstring tendon autograft can provide an earlier return to sports [13]. This situation is continuing to be analyzed prospectively with detailed and high numbers of patient groups in order to see clearer results. In terms of strength, Riesterer et al. [20] conducted a study on pre- and 6-month post-operative isokinetic knee strengths of athletes who underwent ST/G ACLR, and found significant strength increases in both ACL and HK groups at 60°/s angular velocity. However, they observed that the strength increases in the ACL group were greater in the Ex phase and similar to the HK group in the Flx phase. Another study examined the pre- and post-operative isokinetic knee strengths of subjects for whom ST/G and only a single semitendinosus autograft (4ST) were used [22]. It was found that the 4ST group had a better strengthening and recovery compared with the ST/G group; however, the results were not statistically significant. Contrary to our current study, some researchers have not found statistically significant findings between pre-op and post-op strength values in ST/G ACLR groups and HK groups [23,24]. Studies have reported that the differences in these prospectively presented findings may result from pre-operative rehabilitation and strengthening processes conducted in the ACL group in particular [17]. It was also reported that this had a positive effect on post-operative findings in conjunction with the progressive rehabilitation applied for at least 6 months after ACLR [20,25]. Although researchers have reported that the hamstring autograft ACLR methods applied with different types such as ST/G and 4ST produced similar findings regarding strength, recovery and agonist-antagonist balance, they argued that the 4ST technique would protect the gracilis tendon for possible future reconstructions [23]. As a matter of fact, our findings from patients undergoing ST/G and those of other researchers support this [26]. Studies supporting this fact with numerical data have proved that the 4ST technique at an angular velocity of 60°/s revealed a much lower strength deficit in hamstring strength compared with ST/G [27,28]. Similarly, there are also studies reporting that the 4ST technique produced a better recovery in the Flx phase compared with ST/G, although not all of them revealed significant findings [29,30].

The present study did not reveal any negative results regarding strength or agonist-antagonist balance in the 6-month post-operative findings of the ST/G autograft ACLR applied on recreational athletes. The only result which can be considered negative was that, in the post-operative results, the increase in hamstring strength produced lower results compared with the quadriceps. However, this finding was not negatively reflected in H/Q ratios. We believe that this resulted from the fact that semitendinosus and gracilis tendons obtained from the hamstring prolonged the recovery in this muscle group. Although postoperative H/Q ratios were within normal ranges, the decrease in H/Q ratio despite the increased strength is thought to be due to better post-operative quadricep recovery. In addition, this may also have resulted from the fact that the subjects in the present study were tested only 6-month post-operation. Indeed, Roger et al. [22] found that hamstring strength was not fully restored in both the ST/G and 4ST groups in their 2-year follow-up; there was a loss of more than 15% and the patients could not regain their contralateral strength. Another 36-month retrospective study reported similar findings between the ACL group and the contralateral group [31]. Contrary to these studies, the post-operative results of our study revealed that the ACL sides had higher results at angular velocity of 60°/s compared with HK and that findings were similar at angular velocities of 180° and 240°/s. We believe that the different findings, particularly at 60°/s and 240°/s, emerging from our results are produced by strength at low angular velocities and by durability at high angular velocities. Roger et al. [22], in line with our view, reported that these two angular velocities did not reveal muscle recovery in the same sense and that these results had to be supported with scores such as IKDC, Lhysholm and Tegner. We believe that the fact that our study group consisted of individuals actively partaking in sports resulted in higher or similar post-operative results compared with the contralateral group. Finally, the post-operative results of our study showed that, after clinical evaluations, there was no problem in returning to sports (low intensity exercise) for recreational athletes on whom ST/G ACLR was conducted.

The present study had several limitations. Firstly, the subjects were referred to the same physical therapy and rehabilitation specialist but the therapy process was not recorded during the rehabilitation. Secondly, the physical activity levels of the subjects were not known since the branches of sports they were actively engaged in had not been registered. Thirdly, we had a relatively small sample size. Finally, although the knee scores and strength values were evaluated in the present study, post-operative radiological evaluations were not followed-up.

## 5. Conclusions

The pre-operative and 6-month post-operative results of the ST/G ACLR technique applied on recreational athletes, showed that there was a high level of recovery, particularly in quadricep strength, and that the strength increase was lower in the hamstring. In addition, it was found that significant results were obtained only at 60°/s regarding H/Q ratios but this difference was within normal ranges. We believe that similar findings found at high angular velocities (180°/s and 240°/s) resulted from similar endurance levels developed by the quadriceps and hamstring muscle groups before and after ACLR. As a result, it can be argued that the ST/G ACLR method is a feasible method for people with high physical activity levels and athletes. However, the short- and long-term results of ST/G and 4ST ACL reconstruction techniques applied on athletes in particular will reveal more explicit conclusions on the subject. In addition, studies on the athletic population using grafts such as the peronous longus tendon may lead to clearer findings in order not to damage the hamstring after ACL injuries and to accelerate the recovery of both the hamstring and quadriceps.

## Figures and Tables

**Figure 1 jcm-12-00063-f001:**
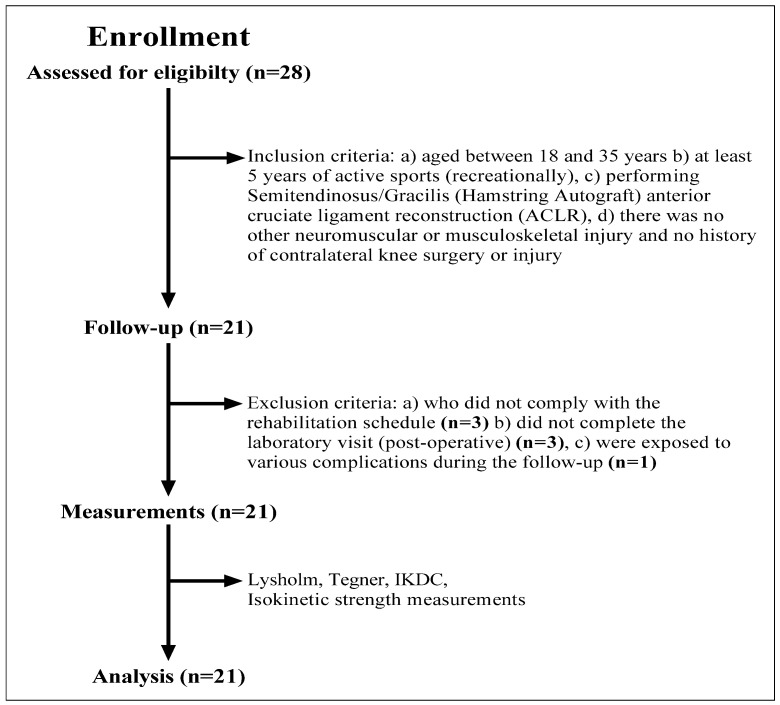
Flow chart diagram: description of the study population and inclusion and exclusion criteria.

**Figure 2 jcm-12-00063-f002:**
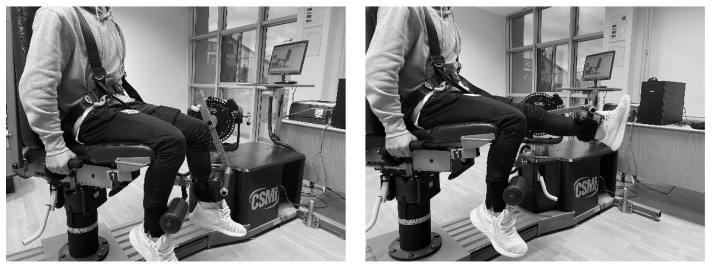
The knee extension and flexion movements by the isokinetic dynamometer.

**Figure 3 jcm-12-00063-f003:**
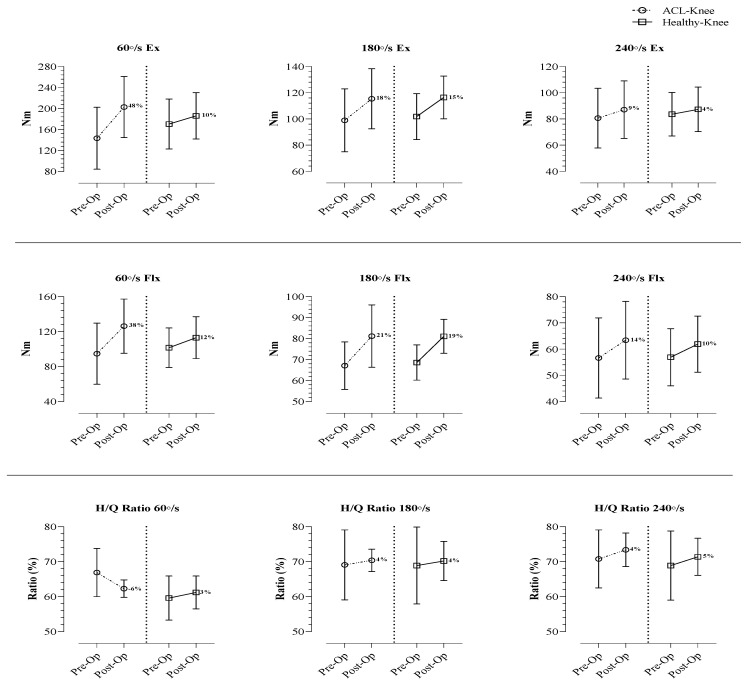
Percentage changes of difference between pre- and post-ACL-Knee (left side of graph) and Healthy-Knee groups (right side of graph).

**Table 1 jcm-12-00063-t001:** Patient demographic characteristics (total 21).

Patients Demographics	Mean ± SD	Median (Min–Max)
Age (years)	25.6 ± 5.8	25 (18–36)
Heigh (cm)	177.9 ± 6.4	177 (170–193)
Weigh (kg)	77.9 ± 13.5	77 (58–114)
BMI (kg/m^2^)	24.6 ± 4.3	24.3 (18.3–35.2)
Follow-up (months)	6.6 ± 0.5	7 (6–7)
Surgical side, n (%)		
*Right*	12 (64.3)	
*Left*	9 (35.7)	

Values are presented as mean ± SD or median (min–max).

**Table 2 jcm-12-00063-t002:** Pre- and post-operative patient-reported outcome scores for Lysholm, IKDC and Tegner.

	Mean ± SD	Median (Min–Max)	95% Confidence Interval		Test Statistical	*p* Value
			Lower	Upper		
Lysholm score						
Pre-operation	73.2 ± 5.8	73.5 (65–84)	−28.439	−21.847	−3.297	0.001 ^1^
Post-operation	98.4 ± 3	100 (90–100)				
IKDC score						
Pre-operation	48.4 ± 9.2	47.5 (36–65)	−45.640	−30.788	−11.117	<0.001 ^2^
Post-operation	86.6 ± 6.7	87 (77–98)				
Tegner score						
Pre-operation	5.8 ± 1.2	6 (4–9)	0.200	0.800	−2.646	0.008 ^1^
Post-operation	5.3 ± 1.1	5 (4–8)				

^1^ Wilcoxon test; ^2^ Paired samples *t*-test; IKDC: International Knee Documentation Committee.

**Table 3 jcm-12-00063-t003:** Knee peak torque variables and the H/Q ratio pre- and post-operative reconstruction in the ACL-Knee and Healthy-Knee groups.

	ACL-Knee			Healthy-Knee			Between Groups		Within Subjects Effects		
	Pre-Op	Post-Op		Pre-Op	Post-Op		Pre-Op	Post-Op	Time	Group	Time × Group
	Mean ± SD	Mean ± SD	*p* Value	Mean ± SD	Mean ± SD	*p* Value	*p* Value	*p* Value	*p* Value	*p* Value	*p* Value
Knee peak torque (Nm)											
60°/s Ex	143.4 ± 59.1	202.9 ± 58.1	<0.001	170.4 ± 47.6	186.1 ± 44.1	<0.001	0.012	0.126	<0.001	0.607	<0.001
180°/s Ex	98.9 ± 24	115.4 ± 22.9	<0.001	101.8 ± 17.6	116.4 ± 16.3	<0.001	0.553	0.856	<0.001	0.697	0.525
240°/s Ex	80.6 ± 22.8	87.1 ± 21.9	<0.001	83.6 ± 16.6	87.3 ± 17	<0.001	0.507	0.972	<0.001	0.709	0.047
60°/s Flx	94.7 ± 35	126.3 ± 35.7	<0.001	101.6 ± 22.6	113.1 ± 23.9	<0.001	0.258	0.044	<0.001	0.588	<0.001
180°/s Flx	67.1 ± 11.3	81.2 ± 14.9	<0.001	68.6 ± 8.4	81.1 ± 8.1	<0.001	0.636	0.984	<0.001	0.828	0.522
240°/s Flx	56.6 ± 15.3	63.4 ± 14.8	<0.001	56.9 ± 10.9	61.9 ± 10.7	0.002	0.946	0.628	<0.001	0.860	0.295
H/Q Ratio (%)											
60°/s	66.9 ± 6.9	62.3 ± 2.5	0.007	59.6 ± 6.3	61.2 ± 4.7	0.014	0.001	0.168	0.091	0.002	<0.001
180°/s	69.1 ± 10	70.4 ± 3.2	0.552	68.9 ± 12.1	70.2 ± 5.6	0.552	0.957	0.901	0.488	0.926	0.975
240°/s	70.8 ± 8.3	73.4 ± 4.8	0.085	68.9 ± 9.9	71.4 ± 5.3	0.162	0.508	0.253	0.054	0.364	0.942

All comparisons had *p*-values ≥ 0.05.

## Data Availability

The datasets used and/or analyzed during the current study are available from the corresponding author on reasonable request.

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
