# Peer review of "Pre- and Post-Operative Hamstring Autograft ACL Reconstruction Isokinetic Knee Strength Assessments of Recreational Athletes"

_jcm, 2022, doi:10.3390/jcm12010063_

Round 1

Reviewer 1 Report

Article is relatively well written, though some sentences are ambiguous. It is well known that the knee flexion strength is reduced after harvesting of ST/G graft. The observation that quadriceps has more power is expected as it is left untouched. The study would have been more meaningful if longer term follow up was available. 

Queries have been incorporated in the pdf file. Please access it by opening the document with adobe reader.

Author Response

Point-by-point response addressing queries from the reviewers

Manuscript ID jcm-2014917 entitled “Pre- and Post-Operative Hamstring Autograft ACL Reconstruction Isokinetic Knee Strength Assessments of Recreational Athletes”

Thank you very much for your valuable contribution to our research. We are ready to do it again and again whenever you have any correction requests. With our best regards

Reviewer 1

Comment #1:

never heard of hard blow to the patella. Infact, blow to the anterior knee can cause posterior knee dislocation and increased risk of PCL injury.

Response #1:

As you said, hard blows to the patella can cause knee dislocation or PCL injuries. However, in the athletic population, hard blows to the patella during active moments may cause injuries to the ACL. However, I am removing this sentence as it is not suitable for you. Thank you for your detailed review.

Comment #2:

Is it a purely retrospective study?

Or were the patients identified retrospectively and followed up prospectively?

If its purely retrospective, then it should be mentioned that data regarding patients were obtained from medical records.

Response #2:

Thank you for reviewing our paper and providing constructive comments. We have done our best to address all your queries. The changes requested in the introduction and discussion section were marked up using the “Track Changes” function.

The study was designed retrospectively. All data collected were part of the routinely ongoing clinical procedure and were derived from medical records. Revised as you said (please see Experimental approach of the study).

Comment #3:

Were the patients prospectively asked to come for the 3 follow-ups?

Response #3:

It was important to monitor the post-operative sensitivity and recovery rates of recreational athletes who are actively engaged in sports. Therefore, we followed the recovery levels by inviting patients to the laboratory outside of routine examinations and recorded these data as part of clinical procedures. Revised the sentence (please see Procedures).

 Comment #4:

Months

Response #4:

Revised as ‘’months’’.

Comment #4:

Already these findings are mentioned in Table 1. I think there is no need of repeating the same thing in the text.

Response #4:

We removed repetitive sentences.

Comment #5:

Eventhough, hamstring and quadriceps strength has improved significantly in the postoperative period, the H/Q ratio has reduced postoperatively (for 60deg/s velocity).

Also, though peak torque of the knee for flexion and extension shows significant improvement postoperatively, but same is not reflected in the H/Q ratio at 60, 180 and 240 deg/s velocity.

How would you justify these observations? Is it purely because the quadriceps has better recovery than hamstrings?

In that case, would you suggest to spare the hamstring and go for another graft like peroneus longus!

Response #5:

Thank you very much for your valuable comment and contribution. As you said, although there is an increase in the extension and flexion forces in the postoperative period, this situation is not the same for the H/Q ratios. As you said, we attribute this to the fact that the hamstring is damaged more and the healing process is better in the quadriceps due to the ST/G method.

In addition, in order not to damage the hamstrings, some researchers argue that the peronous longus muscle can be used as an ACL graft. However, it is not known how the loss of inversion and eversion of the wrist when the peronous longus muscle is taken for the athletic population affects the athletic population, so we think that this decision should be made after extensive research. Although extensive research has been done, we have yet to read such a study.

Thank you very much for your valuable contribution. In line with what you said, we will make some additions to the discussion and conclusion section.

Comment #6:

There seems to be a discontinuity in this sentence. The initial part of the sentence is missing

Response #6:

We revised.

Comment #7:

difference between pre and ...

Response #7:

As you said, we revised.

Comment #8:

rather than telling significant findings were noted repeatedly, mention clearly what was the finding which were noted - improvement or deterioration?

Response #8:

Thank you for your valuable contributions. We tried to revise the findings of our research as you said.

Comment #9:

the last part of this sentence is confusing, especially that "ACL side had higher strength values than HK side both pre and post operatively, but there was no increase in strength postoperatively". Please reframe the sentence to make it more clear. 

Response #9:

Thank you very much for your alert. After re-reading, we realized that the sentence was complex. We tried to revise the sentence to an understandable level.Comment #9:Please refer to table 3. The H/Q ratio for 60deg/s velocity shows a decrease in the ratio postoperatively. Isnt it a negative response which you are validating in the next sentence.

Response #10:

Thank you very much for your detailed review, we wrote a sentence that supports the result as you said and tried to revise it.

Reviewer 2 Report

 More clinical point for orthopedic surgeon:

Compare the strength testing to PROMs, statistical differences?

Which was the used ACLR graft (length, diameter, anatomical or not etc); just for to the physician that he / she can ; its not only which graft you use.

It says retrospective study; if this differs from routine post-operative controls, this seems to be prospective study.

What was the strength device used for testing; figure?

Table 1; give the reader ranges, not standard deviations.

You don’t have to repeat the Table information in text

Author Response

Point-by-point response addressing queries from the reviewers

Manuscript ID jcm-2014917 entitled “Pre- and Post-Operative Hamstring Autograft ACL Reconstruction Isokinetic Knee Strength Assessments of Recreational Athletes”

Thank you very much for your valuable contribution to our research. We are ready to do it again and again whenever you have any correction requests. With our best regards

Reviewer #2

Comment #1:Compare the strength testing to PROMs, statistical differences?

Response #1:

Thank you for your precious comment. Since the data we used in our study were obtained from the data we routinely followed before and after ACLR, we could only evaluate the extension and flexion peak torque ratios and H/Q ratios of the patients. As you said, being able to compare PROM values was important to evaluate the results, but we could not reach these results. In our further research, we will try to use additional calculated parameters such as PROM, joint angle, time to peak torque values in addition to traditional parameters, taking into account what you have said. Thank you very much for your detailed evaluation.

 Comment #2:Which was the used ACLR graft (length, diameter, anatomical or not etc); just for to the physician that he / she can ; its not only which graft you use.

Response #2:

Graft length, diameteri and anatomical information have been added to the surgical procedures section in the method section. Thank you for your valuable contribution.

 Comment #3:It says retrospective study; if this differs from routine post-operative controls, this seems to be prospective study.

Response #3:

It was important to monitor the post-operative sensitivity and recovery rates of recreational athletes who are actively engaged in sports. Therefore, we followed the recovery levels by inviting patients to the laboratory outside of routine examinations and recorded these data as part of clinical procedures. The study was designed retrospectively because all data collected was part of a routinely ongoing clinical procedure and was derived from medical records.   Comment #4:What was the strength device used for testing; figure?

Response #4:

We added the strength device’s figure; thank you for your contribution

  Comment #5:Table 1; give the reader ranges, not standard deviations.

Response #5:

We added to table 1. Comment #6:You don’t have to repeat the Table information in text

Response #6:

We removed repetitive sentences.

Reviewer 3 Report

Many thanks to the authors for having presented a so interesting retrospective study about “Pre- and post-operative hamstring autograft ACL reconstruction isokinetic kee strength asessments of recreational athletes “.

Please before resubmitting the revision version of the article, read the editorial rules carefully, and check for other editorial aspects, such as: text alignment, text justification at the head, etc. The language is so good that the manuscript does not need to be corrected by a person of English mother tongue.

Abstract

The abstract is well structured, it contains the main informations and result of the study. 

Key words

Please provide them in alphabetic order.

Introduction

The introduction is well structured. It well explains the aim of the study.

Lines 41-42: In addition, these mechanical, proprioceptive and efferent neuromuscular disorders resulting from ACL rupture can cause muscle deficits and significant loss in muscle strength [3].

Please, add more details about this injury mechanism and quote also:

·         Hamstring Strain Injury (HSI) Prevention in Professional and Semi-Professional Football Teams: A Systematic Review and Meta-Analysis. Int J Environ Res Public Health. 2021 Aug 4;18(16):8272. doi: 10.3390/ijerph18168272.

Materials and methods

This section contains enough information to understand and possibly repeat the study. It well defines inclusion and exclusion criteria, the sources of the patient, the period of recruitment and follow-up period. It does reflect the Strobe Statement-Checklist for cohort studies. However, the number of patients is relatively small to justify the results. Please, add this aspect in the limits section of the study.

Statistical analysis

Please provide who performed the analysis: an independent statistician or the same authors?

Results

The results presented are quite complete, the number of patients reflect the MM section. The results are reproducible and reflective of clinical expectations. They are displayed in a readable fashion.

Discussion

The length and content of the discussion communicate the main information of the paper. The discussion explains the results relative to prior publications. It well recognizes the limitations of the study. However, the discussion does not explain the results relative to prior publication completely. Please, discuss your result considering also those achieved by other techniques, quoting:

·         ACL reconstruction using a bone patellar tendon bone (BPTB) allograft or a hamstring tendon autograft (GST): a single-center comparative study. Acta Biomed. 2019 Dec 5;90(12-S):109-117. doi: 10.23750/abm.v90i12-S.8973.

Conclusions

The conclusions reflect and refer to the results and the methods of the study.

References

The references are not up to date. However, delate those before 2010 if not essential, replacing them with newer ones and integrating with those suggested previously.

Figures and tables

The number and quality of figures and tables is appropriate to transmit the main information of the paper.

Author Response

Point-by-point response addressing queries from the reviewers

Manuscript ID jcm-2014917 entitled “Pre- and Post-Operative Hamstring Autograft ACL Reconstruction Isokinetic Knee Strength Assessments of Recreational Athletes”

Thank you very much for your valuable contribution to our research. We are ready to do it again and again whenever you have any correction requests. With our best regards

Reviewer #3

Comment #1:

Many thanks to the authors for having presented a so interesting retrospective study about “Pre- and post-operative hamstring autograft ACL reconstruction isokinetic kee strength asessments of recreational athletes “.

Please before resubmitting the revision version of the article, read the editorial rules carefully, and check for other editorial aspects, such as: text alignment, text justification at the head, etc. The language is so good that the manuscript does not need to be corrected by a person of English mother tongue.

Response #1:

Thank you for reviewing our paper and providing constructive comments. We have done our best to address all your queries.

Comment #2:

The abstract is well structured, it contains the main informations and result of the study. The abstract is well structured, it contains the main informations and result of the study. 

Response #2:

Thank you for reviewing our paper and providing constructive comments.

Comment #3:

Key words

Please provide them in alphabetic order.

Response #3:

We revised.

Comment #4:

The introduction is well structured. It well explains the aim of the study.

Lines 41-42: In addition, these mechanical, proprioceptive and efferent neuromuscular disorders resulting from ACL rupture can cause muscle deficits and significant loss in muscle strength [3].

Please, add more details about this injury mechanism and quote also:

Hamstring Strain Injury (HSI) Prevention in Professional and Semi-Professional Football Teams: A Systematic Review and Meta-Analysis. Int J Environ Res Public Health. 2021 Aug 4;18(16):8272. doi: 10.3390/ijerph18168272.

Response #4:

Thank you very much for your contribution. We tried to add it by referring to the research you suggested as you said.

Comment #5:

Materials and methods

This section contains enough information to understand and possibly repeat the study. It well defines inclusion and exclusion criteria, the sources of the patient, the period of recruitment and follow-up period. It does reflect the Strobe Statement-Checklist for cohort studies. However, the number of patients is relatively small to justify the results. Please, add this aspect in the limits section of the study.

Response #5:

Thank you for reviewing our paper and providing constructive comments. As you said, we added as a study limitation the small sample size.

Comment #6:

Please provide who performed the analysis: an independent statistician or the same authors?

Response #6:

As you said, we added the researchers who applied and control the analysis to the "author contributions" section.

Comment #7:

The results presented are quite complete, the number of patients reflect the MM section. The results are reproducible and reflective of clinical expectations. They are displayed in a readable fashion.

Response #7:

Thank you for reviewing our paper and providing constructive comments.

Comment #8:

The length and content of the discussion communicate the main information of the paper. The discussion explains the results relative to prior publications. It well recognizes the limitations of the study. However, the discussion does not explain the results relative to prior publication completely. Please, discuss your result considering also those achieved by other techniques, quoting:

ACL reconstruction using a bone patellar tendon bone (BPTB) allograft or a hamstring tendon autograft (GST): a single-center comparative study. Acta Biomed. 2019 Dec 5;90(12-S):109-117. doi: 10.23750/abm.v90i12-S.8973.

Response #8:

Thank you very much for your valuable contributions. We discussed our postoperative results related to knee scores together with the sources you mentioned in the discussion section and we referred to the sources you mentioned.

Comment #9:

The conclusions reflect and refer to the results and the methods of the study.

Response #9:

Thank you for reviewing our paper and providing constructive comments.

Comment #10:

References

The references are not up to date. However, delate those before 2010 if not essential, replacing them with newer ones and integrating with those suggested previously.

Response #10:

Thank you very much for your valuable suggestions. Yes, some of our resources are unfortunately not very up-to-date, but there is really important information in these resources. And although some of them are very old, they are still up to date. We tried to make some changes to the sources. However, if you insist on changing resources, we will try to change again. We are grateful for your contribution to our research.

Comment #11:

Figures and tables

The number and quality of figures and tables is appropriate to transmit the main information of the paper.

Response #11:

Thank you for reviewing our paper and providing constructive comments.

Round 2

Reviewer 3 Report

The authors answered to my comments properly, improving the quality of their manuscript.

Well done!